# Genetic Basis of Dual Diagnosis: A Review of Genome-Wide Association Studies (GWAS) Focusing on Patients with Mood or Anxiety Disorders and Co-Occurring Alcohol-Use Disorders

**DOI:** 10.3390/diagnostics11061055

**Published:** 2021-06-08

**Authors:** Kaloyan Stoychev, Dancho Dilkov, Elahe Naghavi, Zornitsa Kamburova

**Affiliations:** 1Department of Psychiatry, Medical University Pleven, 5800 Pleven, Bulgaria; 2Department of Psychiatry, Military Medical Academy Sofia, 1606 Sofia, Bulgaria; psihiatria@vma.bg; 3Medical University Pleven, 5800 Pleven, Bulgaria; elahe.97@gmail.com; 4Department of Medical Genetics, Medical University Pleven, 5800 Pleven, Bulgaria; zornicakamburova@gmail.com

**Keywords:** mood disorders, anxiety disorders, alcohol use disorders, comorbidity, genetics, GWAS

## Abstract

(1) Background: Comorbidity between Alcohol Use Disorders (AUD), mood, and anxiety disorders represents a significant health burden, yet its neurobiological underpinnings are elusive. The current paper reviews all genome-wide association studies conducted in the past ten years, sampling patients with AUD and co-occurring mood or anxiety disorder(s). (2) Methods: In keeping with PRISMA guidelines, we searched EMBASE, Medline/PUBMED, and PsycINFO databases (January 2010 to December 2020), including references of enrolled studies. Study selection was based on predefined criteria and data underwent a multistep revision process. (3) Results: 15 studies were included. Some of them explored dual diagnoses phenotypes directly while others employed correlational analysis based on polygenic risk score approach. Their results support the significant overlap of genetic factors involved in AUDs and mood and anxiety disorders. Comorbidity risk seems to be conveyed by genes engaged in neuronal development, connectivity, and signaling although the precise neuronal pathways and mechanisms remain unclear. (4) Conclusion: given that genes associated with complex traits including comorbid clinical presentations are of small effect, and individually responsible for a very low proportion of the total variance, larger samples consisting of multiple refined comorbid combinations and confirmed by re-sequencing approaches will be necessary to disentangle the genetic architecture of dual diagnosis.

## 1. Introduction

Alcohol use disorders (AUDs) affected >100 million people in 2016 [1], whereas, a year earlier, 322 million lived with depression and 264 million suffered from anxiety disorders [2]. In addition to being globally significant health problems in their own right, these disorders occur together much more frequently than expected by chance, confirming a phenomenon known as *comorbidity* or *dual diagnosis* which has been verified by a number of population- [3,4,5,6,7,8,9] and clinically [10,11,12,13,14] based studies over the past three decades. Generally estimated, odds ratios (OR) for the association are 1.64 and 1.53 for the combination anxiety disorder–AUD and depression–AUD respectively [15].

Co-occurring mood and anxiety disorders increase the severity of AUD and associated disability [16] whereas AUD negatively impacts anxiety and mood disorder by worsening symptoms [17], magnifying suicide risk [18], and compromising treatment efficacy [17,19]. Thus, advances in understanding the mechanisms underlying comorbidity will ultimately result in better treatment outcomes. The variety of explanation hypotheses of comorbidity [20] may be broken down to two groups [21]. According to the causal or illness-mediated theories [22], a primary alcohol, mood, or anxiety disorder directly or indirectly causes the secondary condition. The theories on shared etiological factors, on the other hand, assume that one or more common causal factor(s) drive the development of both disorders. While causal theories are outside the scope of this paper, common neurobiological etiology, particularly the research on the genetic basis of comorbidity, will be discussed below in detail.

In the past several decades, twin and other behavioral genetic studies have shown a substantial genetic overlap between internalizing disorders of the mood and anxiety spectrum and externalizing disorders encompassing alcohol and drug dependence [23,24,25]. Subsequently, a more comprehensive exploration of the genetic basis of comorbidity was provided by linkage studies [26] which examine families with multiple affected members to detect chromosomal regions with genetic risk variants [27]. In a landmark linkage mapping study on alcohol–depression comorbidity, Nurnberger et al. [28] reanalyzed dataset from the Collaborative Study on the Genetics of Alcoholism (COGA) [29,30], consisting of 1295 individuals from families affected by alcoholism, confirming a locus on chromosome 1 (near the 120 cm region) containing gene(s) that significantly predispose individuals to alcoholism, depression, or both. Notably, several subsequent studies [31,32,33] have confirmed at least two linkage regions at around 70 and 120 cM of chromosome 1 as substantially associated with neuroticism—a personality trait intimately related to a broad array of anxiety symptoms, depression, and alcoholism [34].

Over the past 15 years, genome-wide association studies (GWAS) have become technically as well as economically affordable and as a consequence of that, they are gradually displacing linkage and other candidate gene studies in the field of psychiatric genetics [35]. GWAS entail screening hundreds of thousands to a million genomic variants (single nucleotide polymorphisms, SNPs) in a case-control design allowing for statistical calculation of each particular variant’s association with the phenotype of interest expressed as an odds ratio of increased or decreased risk [36]. Initially successful with the detection of genes involved in age-related macular degeneration in 2005 [37], the GWAS design has now been broadened to numerous complex traits, including neuropsychiatric illnesses. Spotting a large number of disease-associated risk loci across the human genome for every phenotype of interest, GWAS confirm the pre-existing assumption that rather than single causal genes (according to the Mendelian pattern of inheritance), hundreds to thousands of genetic variants widely spread in the population may confer small accretions of risk for a common disease [38]. In complex genetic disorders such as mental illnesses, the GWAS approach is much more powerful in distinguishing meaningful illness-associated genetic loci as compared to the traditional twin and linkage studies discussed above. While in the most common type of GWAS sampling individuals with schizophrenia or bipolar disorder the attributable risk to each genome-wide significant SNP is small (OR < 1.2), the accumulation of multiple risk SNPs allows for the development of the composite weighted sum of the effects of all common variants, known as the polygenic risk score (PRS) [36]. The latter might be able to aid diagnostics of psychiatric disorders in the near future [39].

Although the current large GWAS databases are focusing on single diagnoses—mostly schizophrenia [36], major depression or bipolar disorder [40], an increasing number of studies in the past 10 years have addressed dual diagnosis phenotypes with a GWAS approach. The current paper attempts to summarize and discuss the results of all the published GWAS exploring samples with alcohol misuse and anxiety or mood disorders comorbidity. In doing so, outlining some possible neurobiological mechanisms and pathways connecting both groups of disorders will be attempted.

## 2. Materials and Methods

We implemented a systematic literature review based on Preferred Reporting Items for Systematic Reviews and Meta-Analyses (PRISMA) guidelines [41].

### 2.1. Search Algorithm

#### 2.1.1. Inclusion Criteria

Articles written in English and published in peer-reviewed journals;Studies performed in humans (animal models relevant to human findings were allowed);Studies of samples with phenotypes of interest—MDD/AUD, BPD/AUD or Anxiety/Anxiety Disorder/AUD identifying the presence of SNPs with a genome-wide level of significance (*p* < 5 × 10^−8^) or suggestive genome-wide level of significance (*p* < 1 × 10^−4^);Papers reporting statistically significant correlation between MDD-PRS, BPD-PRS or Anxiety/Neuroticism PRS and alcohol phenotypes—DSM-IV alcohol dependence or alcohol abuse.

#### 2.1.2. Exclusion Criteria

Studies including alcohol phenotypes that are not based on DSM-IV/5 or ICD-10 criteria, but on screening or other tools for assessment of alcohol use instead—e.g., Alcohol Use Disorders Identification Test (AUDIT) [42].

### 2.2. Data Sources and Keywords

EMBASE, Medline/PUBMED, and PsycINFO databases were searched for a period of 10 years—from 01/01/2010 to 31/12/2020 with the following keywords:” Co-occurring disorders”, “Comorbidity”, “Dual Diagnosis”, “Mood disorder(s)”, “Major Depression (MDD)”, “Bipolar Disorder”, “Anxiety Disorder(s)”, Alcohol Use Disorder”, “Alcohol Abuse”, “Alcohol Dependence” and “Genome-wide association study(ies) (GWAS)”. While analyzing articles identified by this search, all papers indexed in the reference sections were explored and included in the review if eligible.

### 2.3. Selection of Studies

A total of 58 studies were detected by the initial search performed by one author (EN). After the removal of duplicates, 22 articles remained. All of them were reviewed in full text by three authors—K.S. and D.D. (psychiatrists) and Z.K. (specialist in medical genetics) to assess their final eligibility for this paper (Figure 1).

## 3. Results

This review includes 15 articles focusing on GWAS in samples with AUD and co-occurring mood or anxiety disorders. Both a narrative approach and statistical measures (as presented by authors) were used to summarize results which are presented on Table 1. In the studies looking for a correlation between PRS for MDD and BPD and AUD phenotype, the former was obtained from the following discovery samples: Psychiatric Genetic Consortium (PGC) MDD-PRS1 [43] and MDD-PRS2 [44] and PGC BPD-PRS1 [45]. The majority of studies included subjects of White/Caucasian adults of European-American (EA), African-American (AA) or European ancestry.

## 4. Discussion

The present paper aims to summarize studies applying the genome-wide association approach in the search of shared genetic diathesis of mood and anxiety disorders with AUDs. GWAS have brought a massive advance in the understanding of genetic mechanisms that underlie mental disorders and the expansion of insight is being now gradually translated from “pure” diagnoses (i.e., schizophrenia, MDD, etc.) to comorbid phenotypes. In one of the first GWAS analyses of high-risk polymorphisms in BPD and SUD, Johnson et al. (2009) [78] established an overlap of the genetic diathesis for both groups of conditions compatible with the polygenic disorders model. Extracting data from several samples with BPD (*n* = 1461) and SUD (*n* = 400), these authors identified nominally significant SNPs in 69 high-risk genes that were common between the BPD samples and 23 of them were also associated with higher risk of SUD. Some of the spotted high-risk loci have been replicated by later studies included in this review—for instance, SNPs in the COLLA2 gene [46] found to be associated with BPD-AD phenotype, or CDH13 gene involved in the MDD-AD association according to Edwards et al. [49] study. In addition, genes belonging to gene families later found to be associated with AUD-mood disorders comorbidity were also identified in Johnson et al.’s pivotal study—for example semaphorins which are a group of transmembrane proteins engaged in axonal guidance during neural development. An SNP within the semaphorin 3A gene was confirmed in 2017 by Zhou et al. [63] to be involved in MDD-AD comorbidity.

It appears that the majority of significant risk-associated SNPs detected in samples with mood and anxiety disorders and co-occurring AUD are located in genome regions primarily engaged in the processes of neural growth, development, and differentiation as well as in the coding of neurotransmitter receptors and ion channels controlled by them. In this respect, a comparison of the results of GWAS with that of the first-generation genetic studies on alcoholism (linkage and candidate gene studies) which emphasize genes involved in alcohol metabolism—e.g., alcohol-dehydrogenase (ADH) gene cluster [79] or genes coding targets of alcohol pharmacodynamic activity—e.g., GABRA2 (GABA-A receptor subunit α-2) [80] is very characteristic. Indeed, some of these early studies have correctly identified genes that were later found to markedly increase the risk of alcoholism being comorbid with mood or anxiety disorders. For example, the early candidate gene for alcoholism DRD2 (dopamine type 2 receptor) [81] was linked through several SNPs to state and trait levels of anxiety in a Korean sample of AD patients (n = 573) by Joe et al. in 2008 [82], only to be confirmed as a genome-wide significant locus for shared vulnerability to both alcoholism and BPD (Levey et al. 2014 [51] and anxiety disorder-problem alcohol use (Colbert et al. 2020 [74]). Interestingly, this same gene along with ANKK1 was recently confirmed by a meta-analysis of the three largest GWAS on depression as having a key role in MDD [83]. Such a finding supports the significant pleiotropic effects of genes underlying mental disorders and the multifunctional nature or neuronal circuits in which the products of these genes are involved.

Other genetic loci captured by first-generation studies have only shown their role in comorbidity by broadening of the initial phenotype. Thus, in the COGA study previously mentioned [29,30], enriching the phenotype of interest from alcoholism only to alcoholism and ADHD, allowed for a recognition (by a Lod score > 3.0) of a locus on chromosome 2 harboring tachykinin receptor gene (TACR1). Subsequently, this gene which codes a receptor for the Substance P neurotransmitter and modulator peptide, involved in stress response and mood and anxiety regulation, has been confirmed as being implicated in BPD–AD phenotype in the GWAS study of Sharp et al. (2014) [50]. It may be expected, therefore, that future GWAS employing broader phenotype definitions (e.g., BPD + ADHD + AUD), could identify yet other, previously unknown or considered alcoholism “specific” genes, as relevant to AUD–mood and/or anxiety disorder comorbidity.

It should be noted however that some promising genetic regions marked by recent genetic studies in comorbid AUD–mood disorder or AUD–anxiety disorder samples, have not been so far replicated by the genome-wide approach. In the MDD-AUD association for example, Procopio et al. (2013) [84] studied a sample of 333 AD women in Austria of which 51 had a combination of MDD and AD known as Type III alcoholism according to the classification of Lesch et al. [85]. The authors found a significant association of the MDD–alcoholism phenotype with haplotypes (i.e., SNPs) within ADCY5 (type 5 adenylyl cyclase protein gene) on chr. 3, ADCY2 (chr. 5), and ADCY8 (chr. 8) that could discriminate type III alcoholism patients from type I and II. The ADCY trans-membrane protein family is intimately related to the functioning of G-protein coupled receptors and is engaged in procedural learning, synaptic plasticity, and neurodegeneration. In addition to that, it has been linked to the vulnerability to alcohol dependence by previous GWAS studies. [86]. However, no study as yet has replicated ADCY protein family’s relevance to the comorbid phenotype of AD and mood or anxiety disorder. Similarly, in the BPD-alcohol abuse phenotype Mosheva et al. (2019) [87] have recently reported a SNP (rs1034936) within the CACNA1C gene which codes the α1-subunit of the L-type voltage-gated calcium channel and has been implicated in various mental disorders (including MDD and BPD) but also in alcohol effects on CNS. However, not a single GWAS has identified it as directly contributing to BPD–AUD comorbidity so far. In the anxiety disorder(s)–AD phenotype, a recent study by Hodgson et al. (2016) [88] in a sample of 1284 Mexican-Americans from 75 pedigrees reported significant bivariate linkage peaks for alcohol dependence–anxiety at chromosome 9 (9q33.1-q33.2). In addition to hosting rare copy number variants (CNV) that have been linked to autistic spectrum disorders, ADHD, and OCD by a large GWAS [89], this locus also contains the astroactin-2 (ASTN2) and tri-component motif protein 32 (TRIM32). The former encodes the homonymous transmembrane protein, which along with related astroactin-1 (ASTN1) (1q25.2) has a key role in glial-directed neuronal migration during the embryonal formation of the neocortex [89] while the latter is engaged in functional control of dysbindin—a protein intimately involved in the genetics of schizophrenia [90] and influencing glutamate and dopamine signalization [91]. It remains to see whether future GWAS with AUD-anxiety disorders phenotypes will replicate the preliminary significant pleiotropic signals for alcohol dependence–anxiety found in 9q33.1-q33.2 locus.

Further, in the context of causal theories of comorbidity mentioned above (22) it should be noted that some GWAS studies support a causal role of one of the associated disorders on the other. Thus, Polimanti et al. (2019) [71], analyzing large datasets of PGC-MDD-PRS2, PGC-AD-PRS, and UK-Biobank, found evidence for the causal influence of MDD on AD (i.e., mediated pleiotropy) but not the opposite. A possible neurobiological mechanism substantiating such a finding could be an inherited dysfunction of the DRD2 gene (discussed above) translated into lower activity of the D2 receptors resulting in anhedonia and compensatory drug or alcohol consumption. Similarly, in the background of anxiety disorders–alcoholism comorbidity, some of the available GWAS data support the occurrence of alcohol misuse in an attempt to alleviate anxiety compatible with the self-medication hypothesis of comorbidity [92]. For instance, Colbert et al. (2020) in the study reviewed above observed a negative correlation between anxiety and alcohol consumption at chromosome 7:68 562 932-69 806 895 which contains the AUTS2 (autism susceptibility candidate 2 gene). AUTS2, involved in activation of gene transcription as well as in neuronal migration during embryonal development, is an important candidate gene for autism spectrum and intellectual disability disorders; along with being expressed in amygdala and frontal cortex, it also influences alcohol consumption in humans [93]. Besides, its downregulation in Drosophila reduces sensitivity to alcohol, thus possibly increasing consumption [93], while in mice deficient in AUTS2, a decrease in anxiety-related behaviors is evident [94]. Hence, it may be speculated that AUTS2 not only affects anxiety and alcohol consumption in inverse directions, but, as a result of altered function, produces higher anxiety levels and subsequent alcohol misuse induced by the self-medication mechanism. Obviously, to validate or reject the hypothesis of mediated pleiotropy in mood and anxiety disorders comorbid with alcohol misuse, future studies with much larger and refined discovery and target samples will be needed.

Finally, several potential target genes and respective neural mechanisms contributing to comorbidity will be outlined. First in the list is glutamate neurotransmission with its probable role in dual diagnosis supported by recent GWAS implicating the glutamate receptor gene GRIA4 in nicotine dependence–MDD phenotype [95] as well as by the finding that alcohol exposure changes the expression of this and other glutamatergic genes [96]. Besides, impaired NMDA receptor functioning seen in BPD may contribute to the increased tolerance to alcohol resulting in alcohol misuse [97]. Another gene with a high likelihood of contributing to dual diagnosis phenotypes is the α-endommanosidase gene MANEA which, despite its unclarified biological function, has variants found to increase anxiety disorder risk in samples recruited from genetic studies of alcohol and drug dependence [98]. Further in the line are genes participating in circadian clock function such as ARNT, ARNT2, and PER2 which have been implicated in anxiety disorders–alcohol dependence comorbidity [99,100] and D-box binding protein gene (Dbp) supposedly influencing the risk for both bipolar disorder and alcoholism [101].

An inherent limitation of genome-wide association design pertinent to its applicability to co-occurring AUD, mood, and anxiety disorders is the inability to link identified high-risk polymorphisms with meaningful neurobiological pathways, thus paving the way for more successful treatment and prevention strategies. Another major restriction of the currently available GWAS focusing on comorbidity is that discovery samples used for identification of high-risk SNPs are of Western European and North-American ancestry only (see table) which substantially limits the generalization of findings across other populations, given that PRS are very sensitive to ethnic background [102]. This implies that variability in a PRS can be seriously affected by allele frequency differences, divergences in estimated effect sizes, and dissimilarities in population structure across various ethnic groups. For example, the 19 G/C (rs1800883) SNP in the serotonin 5A gene (5-HT5A) was found to have a protective role in relation to BPD risk in a British sample (*n* = 374) [103] whereas this same variant was significantly associated with BPD risk in a Bulgarian candidate gene study (*n* = 450) [104]. Furthermore, PRS usually measures only the contribution of common SNPs in an individual not accounting for other classes of variation which may also influence genetic risk. For instance, copy number variants known to exert a large impact on disease risk are not included in a typical PRS. For the same reason, rare pathogenic alleles are quite often eliminated from PRSs derived from a GWAS summary statistics, because GWASs by definition only include “common” variants with a population prevalence ≥ 1%. Finally, as mentioned by some [49], the discovery samples from which MDD and BPD-PRS are extracted (like for example the PGC-MDD-PRS and PGC-BPD-PRS) might include a high number of alcohol or other substance-induced mood episode cases which significantly confounds the results of studies exploring shared genetic diathesis between alcohol use disorders and mood and anxiety disorders based on PRS.

## 5. Conclusions

In summary, GWAS exploring the genetic background of comorbid AUD, mood, and anxiety disorders demonstrate that multiple genetic variants with different directions and magnitudes influence the development, manifestation, and variation of these dual diagnosis phenotypes. Comorbidity risk is probably conveyed by genes engaged in neuronal development, connectivity, and signaling. It may therefore be hypothesized that comorbidity might represent an expression of a neurodevelopmental disruption that affects cortical and other areas involved in executive functioning, and emotional, and reward processing. In turn, that renders affected individuals susceptible to the occurrence of both mental disorder and SUD, including alcohol [22].

In addition to the intrinsic restriction of GWAS design, a significant barrier to eliciting the role of genetic variants involved in comorbidity is that they supposedly interact with one another (epistasis), may be involved in multiple phenotypes (pleiotropy), and are subject to complex epigenetic influences which are currently largely unknown.

Given that genes associated with complex traits including comorbid clinical presentations are of small effect, and are individually responsible for a very low proportion of total variance, larger samples consisting of multiple refined comorbid combinations and confirmed by re-sequencing approaches will be necessary to disentangle the genetic nature of dual diagnosis.

## Figures and Tables

**Figure 1 diagnostics-11-01055-f001:**
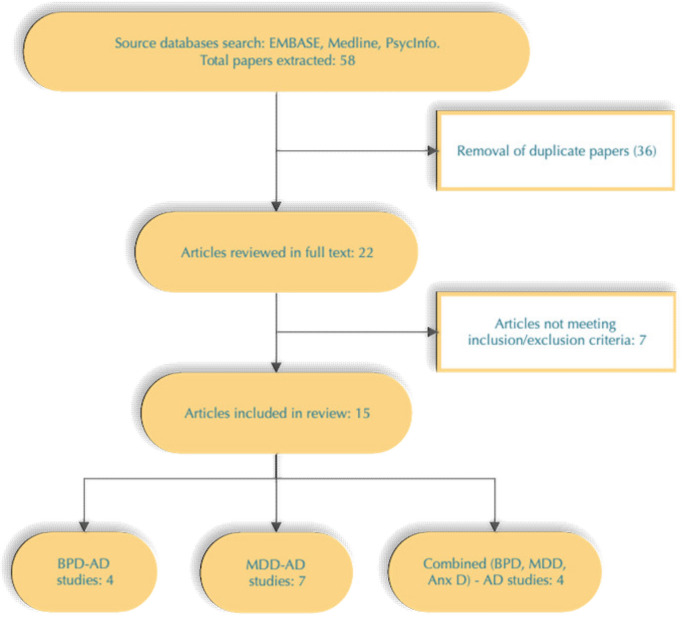
Study selection process.

**Table 1 diagnostics-11-01055-t001:** Overview of GWAS focusing on comorbid mood, anxiety, and alcohol use disorders.

Study	Sample	Identified High-Risk Polymorphisms/PRS Associations	Neurobiological Underpinnings	Comments
Lydall et al. 2011 [46]	506 bipolar I disorder (BPD-I) cases (m = 193) from the University College of London cohort 1 (UCL1) and 510 controls (m = 217). Cases were of English (Caucasian), Irish, Scots and Welsh ancestry. Two phenotypes were defined: ICD-10/DSM-III-R bipolar disorder + RDC ^1^ alcoholism (BPALC)—143 cases (m = 80); Bipolar disorder without alcoholism (NABPD)—363 cases (m = 113).	Suggestive significance (*p* < 1 × 10^−1)^) was detected for the following SNPs, located in or near to genes previously implicated in alcoholism: rs429065 (16q22, *p* = 1.03 × 10^−4^) in the region of CDH11 (cadherin 11 gene); rs3130159 (chr. 6p21.3, *p* = 2.83 × 10^−3^) within COL11A2 (collagen type 11 a2 gene); rs17113138 (5q33.1, *p* = 1.43 × 10^−2^) within NMUR2 (neuromedin U receptor 2 gene); rs7013323 (8p21.3, *p* = 7.99 × 10^−3^) within XPO7 (exportin 7 gene); rs2256569 (5p15.31, *p* = 2.11 × 10^−4^)—SEMA5A (semaphorin-associated protein 5A gene).	Cadherin11—belongs to a group of transmembrane proteins that mediate Ca^++^dependent cell–cell adhesion and the generation of synaptic complexity in the developing brain, implicated in mnemonic processes, addictions, and BPD. COL11A2—encoding one of the two chains of XI collagen; implicated in various facial and skeleton bone dysplasia syndromes as well as Mendelian inherited sensorineural deafness; NMUR2—belongs to the G-protein coupled receptor 1 family and is expressed in guts and CNS (hypothalamus). It binds to the neuropeptide U. The receptor plays a role in food intake and body weight. XPO7 protein—mediates the nuclear export of proteins with broad substrate specificity. Involved in alcoholism according to the pooled COGA genome associated data [47]; SEMA5A—a member of the semaphorins family of membrane proteins involved in axonal guidance during neural development. Associated with autism susceptibility.	Most significant SNPs associations with BPALC phenotype were within or near genes involved in cell adhesion, differentiation and regulation, neurotransmitter pathways and ion function, enzymatic activity, cellular messengers (second messengers), connective tissue. Association between these SNPs and the BPALC cases, but not with NABPD cases suggests genetic effects on alcoholism independent of bipolar affective disorder. Genes of the GABA system (e.g., GABA receptor type 2), which are among the most replicated in alcoholism, were not determined as being associated with BPALC phenotype, indicating either genetic heterogeneity of alcoholism, or the possibility that alcoholism in BPD is mediated by different pathways. Limitations: Small sample for a GWAS.
Kerner et al. 2011 [48]	1000 EA ^2^ subjects (m = 499) with BPD-I (DSM-IV) and 1034 controls (m = 532). In BPD sample, 250 patients (group 1) had lifetime alcohol dependence (AD) with or without lifetime substance abuse/dependence (including nicotine); 40% of another group of 270 patients with BPD with psychotic features (group 2) had a lifetime alcohol abuse but not AD.	The SNP rs2727943 (3p26.3, *p* = 3.36 × 10^−8^) was associated with OR of 4.9 for having BD-I with comorbid alcohol dependence (group 1) phenotype. This polymorphism is located between the genes contactin-4 precursor (BIG-2) and neural adhesion molecule contactin 6 (CNTN6). In group 2, statistical significance on genome-wide level was detected for rs1039002 (6q27.5, *p* = 1.7 × 10^−8^) and rs12563333 (1q41, *p* = 5.9 × 10^−8^). Besides, two SNPs neared significance: rs9493867 (6q23.2, *p*-value from 1.0 × 10^−7^ under recessive model to 9.0 × 10^−8^ under dominant) within the gene encoding serine/threonine kinase (Skg1); and rs13220542 (6q15, *p* = 9.0 × 10^−8^ under dominant model) located 3′ to the gene coding mitogen-activated protein kinase, kinase, kinase 7 (MAP3K7).	The high-risk SNP is located in a region that is deleted in individuals with autistic features. Proteins coded by BIG-2 and CNTN6 might play a role in the formation of axon connections in the developing brain. rs1039002 is located in transcribed genomic sequence with unknown function. The nearest known gene is phosphodiesterase 1 (PDE10A) which is involved in the elimination of intracellular cAMP and cGMP signaling molecules. Inhibitors of the PDE10A have shown therapeutic potential in Parkinson’s and Huntington’s disease, addiction, and OCD and are being tested in clinical trials. Rs12563333 is located in a transcribed sequence immediately upstream of the gene MAP/microtubule-affinity regulating kinase 1 (MARK1). MARK1 phosphorylates microtubules associated proteins and is involved in synaptic plasticity and dendritic trafficking. The two SNPs closing genome- wide significance are located in genes involved in response to stress through K, Na, and Cl channels (Skg1) and activation of protein kinases such as MAPK8 and MAP2K4 (MAP3K7).	The study distinguished three distinct profiles of comorbidity in the BP-1 sample with two of them significantly associated with specific SNP/SNPs: a group with comorbid psychosis and substance abuse (including alcohol abuse but no alcohol dependence); a group with comorbid alcohol dependence but also high lifetime prevalence of comorbid PD; and a group with a very low rate of co-morbid conditions. This suggests that phenotype heterogeneity in BPD might indicate genetic heterogeneity. The SNPs close to genome-wide significance are in genes implicated in stress response and warrant further investigation in samples with BD comorbid with SUD. Since the associated variants were rare, future studies applying re-sequencing of these chromosomal regions in BP patients could be more appropriate for replication. Limitations: Small sample for a GWAS.
Edwards et al. 2012 [49]	467 cases with DSM-IV AD and major depression (MDD) phenotype (m = 287) and 407 unaffected controls (m = 132). Cases were drawn from the COGA study sample [29] and were from EA and AA ^3^ descent (so were controls).	No marker met genome-wide significance criteria (5 × 10^−8^); 10 SNPs had *p* values < 1 × 10^−5^ and 7 of them fell into the regions of the known genes: OXTR (oxytocin receptor gene), FAF1 (Fas-associated factor 1), OPA3 (Optic atrophy 3), EFHA2 (EF-hand domain family, member 2), FHIT (fragile histidine triad gene), WDR7 (WD repeat domain 7), SPATA13 (spermatogenesis associated 13). A number of SNPs with *p*- value < 1 × 10^−3^ were detected in glutamate receptor genes (GRIN2A, GRIN2C, and GRID1) which have been previously associated with SCH, MDD, and addiction; as well as within genes previously associated with depression, AD, or other addictions (CDH13, CSMD2, and HTR1B),	FAF1, OPA, EFHA2, and WDR7 genes encode protein products engaged in apoptosis (among their other functions) FHIT’s coded enzyme is involved in purine metabolism, but also in protection against DNA damage SPATA13 encodes a protein involved in cell migration, adhesion, assembly and disassembly. CDH13—a member of the cadherin family of cell adhesion molecules, impacts GABA functioning and is a risk gene for ADHD, SUD, MDD, and violent behavior; CSMD2—codes a synaptic transmembrane protein involved in the development and maintenance of dendrites and synapses that has been linked to schizophrenia and autistic disorders by GWAS. HTR1B—codes 5-HT receptor 1B receptor associated with OCD, personality disorders, and schizophrenia.	The degree of overlap of significant SNPs between a comorbid phenotype (AD-MDD) and an AD-only phenotype is modest suggesting that comorbid phenotype is partially influenced by genetic variants that do not affect AD alone. Limitations: small sample size for GWAS; >50% of cases have not met DSM-IV criteria for independent MDD, i.e., depressive symptoms in them have occurred under the influence of alcohol or drugs
Sharp et al. (2014) [50]	The sample consisted of 2096 patients and 1056 controls. Patients were distributed as follows: 506 BPD-I cases from UCL1; 593 cases (m = 219) from the University College of London cohort 2 (UCL2), of which 409 were with BPD-I and 184 with BPD-II; 997 AD syndrome cases from the University College of London ADS sample (UCL ADS) part of the UK-COGA project; 35 cases of ADHD ^4^Cases were of English (Caucasian), Irish, Scots, and Welsh ancestry. Control subjects were comprised of 672 screened individuals without mental disorder or family history for schizophrenia, AD, BPALC, and 384 unscreened ones.	Two SNPs in tachykinin receptor 1 gene (TACR1, 2p12) were significantly associated with BPD cases in comparison with screened controls—rs3771829 (*p* = 0.002, OR 1.57) and rs3771833 (*p* = 0.004, OR 1.43). However, neither of the two were associated with BPD in the UCL2 sample alone. In comparison with controls, rs3771829 was significantly associated with BPD (UCL1 and UCL2 combined, *p* = 9.0 × 10^−8^ under dominant model), ADS (*p* = 2.0 × 10^−3^) and BPALC (*p* = 6.0 × 10^−4^). DNA sequencing in selected cases of BPD and ADHD with inherited TACR1-susceptibility haplotypes determined 19 SNPs in different regions of TACR1 that increase vulnerability to BPD, ADS, ADHD, and BPALC. The association with TACR1 and BPAD, ADS, and ADHD suggests a shared molecular pathophysiology between these disorders.	Neurokinin 1 receptors (NK1R) encoded by TACR1 are abundantly expressed throughout brain regions driving reward and reinforcement. The binding density of NK1R is highest in the locus coeruleus involved in mood regulation and response to stress. Inactivation of NK1Rs critically modulates alcohol reward and escalation, supporting a direct role of NK1R in the regulation of alcohol intake and the development of alcohol dependence. NK1Rs are an attractive molecular target for the treatment of alcohol use disorders but also depression and anxiety. Trials of the efficacy of NK1R antagonists in ADS are currently underway.	The lack of association of the two top marker SNPs with BPD in UCL2 sample may reflect both the heterogeneity of BPD susceptibility genes even in single ancestrally originating cases, as well as the presence of low frequency disease alleles. Differences of association of rs3771829 and rs3771833 was stronger for BPALC only compared to screened controls than for BPD-total compared to screened controls. Therefore, it is likely that the comorbid ADS in BPD cohort is driving the association, i.e., NK1Rs are more strongly implicated in the neurobiology of alcohol use disorders than in BPD. Limitations: Small sample for a GWAS.
Levey et al. 2014 [51]	7948 subjects (4519 patients and 3429 controls). Patients were distributed as follows: 1151 men from German-Caucasian descent with AD [52,53]; 2768 patients (m = 1687) from EA (*n* = 1273) and AA (*n* = 1495) origin with AD; 600 patients (m = 366) from EA (324) and AA (*n* = 276) origin with alcohol abuse. Controls were distributed as follows: 2168 subjects form German-Caucasian descent (m = 939); 1261 subjects (m = 475) from EA (388) and AA (873) descent.	Authors used a translational Convergent Functional Genomics (CFG) approach to discover genes involved in alcoholism by integration of GWAS data with other genetic and gene-expression data from human and animal model studies. Top 11 candidate genes for alcoholism detected by this study (*p* = 0.041) were explored for the degree of overlap with risk genes for BPD and anxiety disorders in previous studies of the same group with a similar design [54,55]. Besides, the association of Genetic Risk Predictive Panel for BPD (i.e., list of top risk SNPs in 56 genes involved in BPD) identified by the same authors in a previous study [54] was tested in AD, AA, and Control samples of the current study. Results: SNPs in SNCA (rs17015888) and DRD2 were shared among anxiety disorder and alcoholism samples while GNAI1, GRM3 (rs17160519 to rs4236502), and MBP genes showed involvement in BPD and alcoholism. Genetic risk prediction score for BP showed increased genetic load for bipolar disorder in both alcohol dependence (*p* = 9.94 × 10^−8^) and alcohol abuse (*p* = 1.18 × 10^−4^).	SNCA (synuclein alpha), a pre-synaptic chaperone, has been reported previously as being involved in modulating brain plasticity and neurogenesis, as well as neurotransmission, primarily as a brake. On the pathological side, low levels of SNCA might offer less protection against oxidative stress, whereas high levels of SNCA may have a role in neurodegenerative diseases, like Parkinson’s disease. SNCA has been described as a susceptibility gene for alcohol cravings and response to alcohol cues. DRD2 (dopamine 2 receptor gene) has shown reduced expression in the brains of alcoholics and one possible explanation for this, bridging the common role of this receptor in AUD and BPD, is that both conditions include hyperdopaminergic state which drives individuals to hedonistic activities and leads to homeostatic downregulation of their DRD2 receptors. An alternative hypothesis sees lower levels of dopamine receptors as a reflection of reduced dopaminergic signaling and anhedonia, leading individuals to overcompensate by alcohol and drug abuse. GRM3 belongs to the metabotropic glutamate receptors family (G protein-coupled receptors), i.e., it is heavily involved in neurotransmitter signaling. GRM3 has been exclusively associated with BPD so far. MBP (Myelin Basic Protein gene) encodes a major constituent of the myelin sheath of oligodendrocytes and Schwann cells GNAI1 (Guanine Nucleotide-Binding Protein G(I) Subunit Alpha-1 gene) encodes a protein that is part of a complex that responds to beta-adrenergic signals by inhibiting adenylate cyclase.	The GWAS study on which discovery was based contained males as probands and males and females as controls. Therefore, it is possible that some of the nominally significant SNPs detected have to do with gender differences rather than with alcoholism per se, or at least, are limited to male alcoholism. Stratification across gender and ethnicities may have also been a confounding factor in US samples. Possible ethnicity differences in alleles, genes, and the consequent neurobiology need to be explored in further larger sample studies, taking into account environmental and cultural factors. Limitations: Small sample for a GWAS.
Carey et al. 2016 [56]	1160 cases and 1413 controls (f = 56%) included in SAGE [57] and sampled from three previous studies—COGA [29], COGEND [58], and FSCD [59]. Cases were of non-Hispanic EA ancestry. In addition to meeting DSM-IV AD criteria, they often met criteria for cocaine, cannabis, and opioid dependence. A measure of general substance involvement (GENSUB) was generated by factor analysis of the individual substance involvement measures (types of substances and frequency of use). Associations between PGC-PRS-MDD1 and PGC-PRS-BPD1 and involvement in AD were tested. PRS-MDD1 includes >200 SNPs related to >180 genes associated with synaptic function and neurotransmission and especially expressed in prefrontal brain areas. BP-PRS1 includes SNPs within or near genes implicated in cell adhesion and migration as well as coding of calcium and other ion channels, neurotransmitter receptors, and synaptic components.	MDD-PRS/Alcohol Dependence: suggestive significance (*p* < 0.0001, OR 1.23, 95%CI) was found for associated MDD-PRS and severe alcohol dependence (6–7 dependence symptoms), supporting shared genetic liability to MDD and AD. Current MDD-PRS explain roughly 1% of the variance in general substance involvement (GENSUB). BPD-PRS/Alcohol Dependence: there was evidence for a dose-dependent relationship between BPD-PRS and an increasing number of alcohol dependence symptoms among regular drinkers with at least one symptom of dependence. Association of BPD-PRS with GENSUB (i.e., involvement in multiple substances) was much stronger than that for alcohol alone.	MDD/Alcohol Dependence: together with data from previous GWAS showing significant overlapping regions/variants contributing specifically to MDD alone, MDD with a comorbid SUD, or a combined MDD and SUD phenotype only, these results suggest that relationships between MDD and alcohol and MDD and other substances (cocaine) are substance-specific. The overall association with general substance involvement liability may be reflective of similar cognitive mechanisms (e.g., impulsivity, emotion dysregulation, sensation- seeking) that are thought to broadly underlie both BPD and substance use disorders. Such a mechanism fits well with the hypothesis for a genetic basis of the BPD-SUD comorbidity supported by many studies.	Limitations: 1. Small sample for a GWAS. Nominal associations may strengthen with larger discovery samples (e.g., samples from which PRS are extracted), as well as larger target samples. 2. While the study confirms that shared genetic architecture contributes to mood disorders and substance use disorders, it does not reveal specific biological (e.g., reward-related neural responsiveness, epigenetically driven gene expression changes), psychological (e.g., anhedonia, impulsivity), and/or experiential (e.g., early life stress, peer group pressure) mechanisms through which risk is manifested.
Andersen et al. 2017 [60]	3871 DSM-IV-AD cases (m = 2551) and 3347 controls (m = 2082) from four different study samples—COGA [29], SAGE [57], Yale-Penn [61], and NHRVS [62]. Subjects were of European-American ancestry. The prevalence of MDD among AD patients in the different samples was between 19 and 35% and for controls—between 6.3 and 12.5%. Associations between PGC-MDD-PRS1 and AD were performed with the analysis corrected for age, sex, and population stratification.	A significant association was observed between MDD-PRS and AD case-control status for all four AD samples (*p* = 3.3 × 10^−9^; *p* value threshold = 0.4). The proportion of variance in AD explained by the MDD-PRS was small (*R^2^* value of 0.0018 (min.) and 0.026 (max). Association remained even when recalculated MDD-PRS from GWAS-MDD samples without comorbid MDD-AD cases was used in analyses performed only for those patients from the four samples with pure AD (i.e., without MDD), providing further support for the genetic overlap between MDD and AD. No difference in the strength or significance or associations between MDD-PRS and AD status by sex was observed.	Although studies like the current one cannot, due to their design, determine the mechanisms by which shared genetic liability for MDD and AD operate, there are some suggestive possibilities that should be tested by future studies. For example, anxiety may be a significant factor linking AD and MDD via the internalizing pathway. Furthermore, broader personality traits such as neuroticism, disinhibition, and sensation seeking are potentially associated with a range of internalizing and externalizing psychiatric disorders, including comorbidity of MDD and AD.	Limitations: 1. MDD-GWAS with larger sample sizes will likely improve the predictive ability of MDD PRS and probably lead to refinement of observed associations. 2. The study included only AD case (and not other SUDs) and for this reason further studies are needed to check whether the MDD-PRS association is specific to AD or it generalizes to substance dependence broadly as suggested by existing research data.
Zhou et al. 2017 [63]	7822 subjects (m = 4480) from EA (3169) and AA (4653) descent from the Yale-Penn Study [61] with lifetime DSM-IV AD and MDD diagnosis. The participants were divided into Yale-Penn 1 and Yale-Penn 2 subsamples based on the period of recruitment (between 1999 and 2015) and on the genotyping platform used.	The SNP rs139438618 at the SEMA3A (semaphorin 3A) gene locus was significantly associated with AD and MD comorbidity in AA participants in the Yale-Penn 1 (β = 0.89; *p* = 2.76 × 10^−8^) and Yale-Penn 2 (β = 0.83; *p* = 2.06 × 10^−4^) There was no significant association identified in EA participants. Analyses of PRS showed that individuals with a higher risk of neuroticism or depressive symptoms and a lower level of subjective well-being and educational attainment had a higher level of AD and MD comorbidity, while larger intracranial and smaller putamen volumes were associated with higher risks of AD and MD comorbidity.	Rs139438618 is located in the intron part of the SEMA3A gene (7q21.11), which codes the homonymous protein part of the semaphorin family. The latter consists of transmembrane and secretion proteins involved in the axonal growth and connectivity acting like chemorepulsors (inhibitors of axonal sprouts) or chemoattractants (stimulators of apical dendrites). The expression of these genes is most intensive in early fetal development in the olfactory brain and cerebral and entorhinal cortex. Previous studies have confirmed their role in schizophrenia, Alzheimer’s disease, epilepsy, and amyotrophic lateral sclerosis as well as intestinal malformations (Hirschprung disease). AD-MDD phenotype was associated with neuroticism PRS including 11 significant SNPs on chromosomes 3, 8, 9,11,15,17,18. One of them, on chromosome 8, is in the zone of MSRA and MTMR9 genes which are both expressed in CNS and code products engaged in repair of oxidatively damaged proteins (MSRA) and cell proliferation control (MTRM9). Both of them are significantly associated with depression/neuroticism and low subjective well-being according to a large GWAS (n = 170,000) [64]. In addition to that, MTRM9 has also been linked to generalized epilepsy with febrile seizures.	The rs139438618 SNP has not so far been identified as risk associated in GWAS studies with pure MDD and pure AD phenotypes, which suggests a pleiotropic effect on the level of a single gene. The presence of this SNP AA only is likely to represent a populational genetic effect.
Reginsson et al. 2018 [65]	8701 cases (f = 32.7%) of alcohol use disorder (DSM-IIIR and DSM-IV) were part of a larger sample (n = 144,609) of Icelandic subjects, including 10,036 individuals admitted for in-patient addiction treatment, 35,754 smokers, and a group of patients with schizophrenia (n = 600) and BPD (n = 772). PGC-BPD1 PRS was tested for association with alcohol dependence.	Higher BPD1-PRS was associated with increased risk of alcohol use disorder (*p* = 1.7 × 10^−9^) and with earlier onset of substance use problems (including alcohol) (OR = 1.16, *p* = 1.9 h 10^−3^). Only alcohol use disorder (and not smoking and other substance use disorders) was nominally associated (OR = 1.09, R2 = 0.59%, *p* = 2.7 × 10^−3^) with BPD-PRS when including PGC-SCZ-PRS [66] as a covariate. This implies that alcoholism may share common genetic causal factors with BPD to a larger extent than smoking and other substance use disorders do.	The results support the notion of common genetic roots of the comorbidity between addiction (including alcohol addiction) and severe mental disorders such as BPD and schizophrenia, as opposed to solely being a direct consequence.	
Muench et al. 2018 [67]	BOLD fMRI ^5^ sample of 45 DSM-IV-AD cases with mean age (m = 35), 45 controls (m = 22) scanned during MID ^6^ task-directed on winning money or avoiding money loss. Subjects were of AA, EA, Asian, multiracial, and unknown ancestry. 12.4% of patients and 1.1% of controls had lifetime anxiety disorders, while current anxiety disorders measures were detected in 9.0% of patients and 1.1 of healthy controls. For mood disorders the corresponding numbers were 11.2% and 9% (lifetime) and 5.6% and 0% (current). NIAAA ^7^ sample of 1123 AD cases (m = 323), 735 controls (m = 325). NIAAA subsample of 955 subjects with lifetime AD (669 males). Subjects were of AA (1178), EA (1383), Asian (68), Multiracial (59), Native American/Alaska, Hawaiian/Pacific (15), and unknown ancestry (103). 5.8% of the patients and 0.4% of the controls met current MDD criteria, while 11.6% of patients and 26% of controls had a lifetime MDD. SAGE sample [57] with 1848 AD cases (1162 EA, 685 AA, males = 60%), and 1990 controls (1346 EA, 644 AA, males = 30 and 36%).	Suggestive significance (*p* = 0.09) was found for the previously associated with MDD risk variant rs10514299 within TMEM161B- MEF2C gene cluster containing Transmembrane Protein 161B gene and myocyte enhancer factor 2C gene. Carrying the minor T allele (TT/CT and not CC) was associated with a lifetime diagnosis of AD (odds ratio = 0.82, *p* = 0.09) in the NIAAA sample. The T allele of rs10514299 was significantly associated with greater depression symptom severity in individuals with a lifetime AD diagnosis (β = 1.25, *p* = 0.02) in the NIAAA sample with this finding driven by individuals of AA ancestry.	TMEM161B’s function is unclear, with gene ontology annotation related to it include nucleic acid binding. MEF2C encodes a transcription factor that has been so far associated with epilepsy and intellectual disability. Carrying the T allele in rs10514299 was associated with a significant increase in putamen activation during high and low loss anticipation in patients with AD, but with a significant decrease in the controls, indicating that the allele differentially affects this neural phenotype in AD. Hence, MDD risk variant rs10514299 in TMEM161B- MEF2C gene cluster was shown for the first time to predict neuronal correlates of reward processing in an AD phenotype implying possible eligibility of this polymorphism as a biomarker for disrupted reward processing in AD individuals. The fact that a MDD risk variant was also shown to be relevant to AD phenotype supports a potential role of the respective genetic locus in an endophenotype related to deficit of reward processing (i.e., anhedonia).	Limitations: 1. No correction for multiple comparisons was done, therefore future confirmatory analyses are needed to validate the functional relevance of rs10514299. 2. Insufficient sample size to detect firmly the likely small effect size of this SNP. In co-occurrence of anxiety disorders for example (GAD, panic disorder and phobias), studies reporting suggestively shared genetic susceptibility loci have employed much larger samples [68].
Foo et al. 2018 [69]	Target sample AD: 1333 German-Caucasian male DSM-IV-AD cases and 1307 German-Caucasian controls from both sexes. A subset of the AD cases (*n* = 332) was recruited explicitly excluding comorbid MDD. Target sample MDD: 597 cases and 1292 controls from German-Caucasian ancestry (52,53). Discovery samples: PGC-PRS-MDD1 (8148 cases, 7955 controls); PGC-PRS-MDD2 (59,265 cases, 112,092 controls).	Significant associations were found between AD disease status and both PGC-PRS-MDD2 (*p*-threshold = 1.0, *p* = 0.00063, *R^2^* = 0.533%) and PGC-PRS-MDD1 (*p*-threshold = 0.2, *p* = 0.00014, *R^2^* = 0.663%) with the larger sample of PGC-MDD2 not building on additional predictive power. In the MDD target sample however, calculating PRS yielded more power with the bigger sample of the PGC-PRS-MDD2 (*p*-threshold = 1.0, *p* = 0.000038, R^2^ = 1.34%) versus PGC-PRS-MDD1 (*p*-threshold = 1.0, *p* = 0.0013, R^2^ = 0.81%). When calculating PGC-PRS-MDD2, PRS in the subsample of AD patients without comorbid MDD, significant associations were still found (*p*-threshold = 1.0, *p* = 0.042, *R^2^* = 0.398%).	The presence of an association between AD disease status and PRS-MDD in a subsample of AD cases without comorbid MDD supports the hypothesis for a substantial genetic overlap between AD and MDD. Although PRS association studies like the present one do not, for the reason of design, possess the power to predict suggestive neurobiological pathways explaining comorbid phenotypes, the authors hypothesized that the level and risk of AD and MDD comorbidity may be linked to neuropsychiatric traits and brain volumes.	Limitations: determining shared genetic etiology in comorbid disorders is inevitably facing the problem of “enrichment” of the comorbid disorders in discovery and target samples. In the current study, there was no information regarding the AD comorbidity of the PGC-PRS-MDD2 sample. For that reason, future studies should employ rigorous phenotyping and improved characterization of samples with particular detailed assessment of comorbidity, symptomatology, and severity.
Walters et al. 2018 [70]	14,904 individuals with DSM-IV-AD and 37,944 controls from 28 case-control and family-based studies conducted in USA, Europe and Australia. Data were stratified by genetic ancestry (European, *n* = 46,568; African, *n* = 6280).	Independent, genome-wide significant effects of different *Aldehyde Dehydrogenase 1B* (ADH1B) gene variants were identified in European (rs1229984; *p* = 9.8 × 10^−13^) and African ancestries (rs2066702; *p* = 2.2 × 10^−9^) Significant genetic correlations were observed between AD and 17 phenotypes in unrelated European samples (10,206 AD cases and 28,480 controls), including neuroticism (*p* = 2 × 10^−6^), depressive symptoms (*p* = 3 × 10^−7^), and MDD (*p* = 3 × 10^−11^).	Given the stringent criteria for patient selection in the study sample (i.e., all individuals were with confirmed AD diagnosis and not less severe forms of alcohol misuse), MDD may primarily share genetic liability with alcohol use at pathological levels and on a molecular level, pleiotropic effects may be implicated.	There is a continuing need to characterize the genetic architecture of AD in non-EU populations and test the genetic correlation between this phenotype and mood/anxiety phenotypes in individuals from non-European ancestry. Larger future samples will allow us to uncover additional pleiotropy between pathological and non-pathological alcohol use, as well as between AD and other neuropsychiatric disorders.
Polimanti et al. 2019 [71]	PGC-MDD2 sample (135,458 cases and 344,901 controls); PGC-AD sample 10,206 cases and 28,480 controls; UK Biobank sample [72] 428,308 individuals from white British ancestry. Four phenotypes were defined: MDD, AD, Alcohol Consumption—Frequency (AC-f), and Alcohol Consumption—Quantity (AC-q).	Linkage disequilibrium score regression and Mendelian randomization (MR) showed a positive genetic correlation between MD and AD. AC-q demonstrated a positive correlation with both AD and MD, while AC-f had a negative correlation with MDD and non-significant with AD. MR analyses confirmed the presence of pleiotropy among these four traits. However, the MD-AD results reflect a mediated pleiotropy mechanism (i.e., causal relationship) with an effect of MD on AD, while there was no evidence for reverse causation.	The study supports a causal role for genetic liability of MD on AD based on genetic datasets including thousands of individuals.	Larger AD and MD datasets will be required to confirm the study findings using genetic instruments based on genetic variants that reached the more conservative genome-wide significance (i.e., *p* < 5 × 10^−8^ for a particular SNP).
Martínez-Magaña et al. 2019 [73]	192 individuals of Mexican Ancestry (125 cases and 67 controls). 72 of the cases had e lifetime DSM-IV schizophrenia (SCZ), while 53 (m = 25) were with BPD diagnosis. Of the latter, 23 had AD or alcohol abuse (in some combined with other SUD—nicotine, cocaine, cannabis, inhalants, or stimulants). Correlational testing of the variance of dual diagnosis (DD ^8^) phenotype explained by PGC-MDD1-PRS, PGC-BD1-PRS, and PGC-SCZ-PRS was performed, i.e., the hypothesis of whether the current PRS might correlate with a lifetime DD was checked.	PGC-MDD1-PRS showed a significant shared genetic etiology with the DD phenotype (Nagelkerke Pseudo-R2 = 0.0451, corr. *p* = 0.0118, n = 334 SNPs) whereas BPD-PRS did not (*p* = 0.1585). Patients with DD in the BD group had a higher MDD-PRS when compared to non-DD BD patients (*p* < 0.05). Notably, PGC-SZC-PRS [66] also demonstrated statistically significant common genetic background with DD phenotype, including DD-BPD (Pseudo-R2 = 0.0283, corr. *p* = 0.0118, n = 8058 SNPs), but it could not discriminate statistically DD-BD cases from non-DD-BD ones. Besides, MDD-PRS explained a higher amount of variance (4.51%) predicting placement in the DD group (for both SCZ and BPD patients) than did the SCZ:PRS (2.83%).	The study results suggest that both the MDD-PRS and the SCZ-PRS might be useful in detecting DD risk. However, when PRSs are applied to a specific diagnosis, MDD-PRS used in patients with BD is the only specific PRS which discriminates DD from non-DD cases. The shared genetic susceptibility between MDD and AD (or alcohol abuse) might drive this result given the fact that the main problem substance in the studied sample was (apart from nicotine) alcohol.	The study is one of the first approximations on how to apply psychiatric PRS in admixed populations (i.e., with ancestry different from European, European-American, or African-American). The application of PRS in different populations, with distinct admixtures and diverse phenotypes, could give more information on the use of PRS for psychiatric disorders as a translational risk prediction. Limitations: small sample size.
Colbert et al. 2020 [74]	>900,000 subjects (cases and controls) from five different GWAS samples with anxiety disorders [68,75], AD [70], problem alcohol use (PAU) [76], and alcohol consumption (AC) [77] phenotypes.	All anxiety phenotypes showed a significant positive genetic correlation with AUDIT-*p* (items of AUDIT associated with problem alcohol use) and AD (r_g_ ≥ 0.35). However, the anxiety phenotypes were uncorrelated with AUDIT-C (part of AUDIT-related AC) and drinks per week, indicating AC was not genetically related to anxiety. In females, three significant positive genetic correlations were found between PAU and anxiety phenotypes DSM-V-like GAD ^9^ and any anxiety disorder. No such correlation was observed in males. 47 independent loci with significant (*p* < 7.60 × 10^−7^) local genetic covariance between pairs of traits were identified and three of them showed positive local genetic covariance between PAU and anxiety phenotypes.	One of the identified loci is at chr. 11 (11:113 105 405-113 958 177) and it contains the dopaminergic pathway gene (DRD2) which substantially moderates stress-induced alcohol consumption in mice and also influences connectivity between basal ganglia and frontal cortices. The region also contains *NCAM* (Neural Cell Adhesion Molecule gene), *TTC12* (Tetratricopeptide Repeat Domain 12 gene) and *ANKK1* (Ankyrin Repeat and Protein Kinase Domain-Containing Protein 1 gene) which, together with DRD2, form the so-called NTAD gene suggestively contributing to various psychiatric disorders as well as the comorbidity of psychiatric disorders. A locus on chromosome 9 (10 879 18811 616 822), previously not implicated in AUD, was also found to have multiple significant positive covariances between anxiety and PAU, but not AC. This locus has been previously associated with worry and neuroticism, depression, and anxiety but has not been associated with alcohol misuse anxiety comorbidity.	Genetic covariance between anxiety traits and PAU is concentrated in certain brain areas: amygdala, caudate basal ganglia and frontal cortex. These results align findings from fMRI studies pointing to the role of these regions in anxiety and alcohol use. Limitations: GWAS sample sizes are not large enough given the small variations of risk associated with identified loci. Limited panels for expression is another limitation of the study. Finally, the analyses does not identify specific mechanisms which contribute to the comorbidity of the two disorders or a causal direction.

^1^ RDS = Research Diagnostic Criteria; ^2^ EA = European-American; ^3^ AA = African-American; ^4^ ADHD = Attention Deficit Hyperactivity Disorder; ^5^ BOLD fMRI = Blood Oxygen Level Dependent Functional Magnetic Resonance Imaging; ^6^ MID = Monetary Incentive Delay; ^7^ NIAAA = National Institute on Alcohol Abuse and Alcoholism; ^8^ DD = Dual Diagnosis; ^9^ GAD = Generalized Anxiety Disorder.

## Data Availability

Not applicable.

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
