# Peer review of "Genetic Basis of Dual Diagnosis: A Review of Genome-Wide Association Studies (GWAS) Focusing on Patients with Mood or Anxiety Disorders and Co-Occurring Alcohol-Use Disorders"

_diagnostics, 2021, doi:10.3390/diagnostics11061055_

Round 1
Reviewer 1 Report
Well performed and clearly written article.
Author Response
Dear Reviewer,
Thank you for your feedback. The suggested changes will be incorporated into the text accordingly. You can find a cover letter in the attachment. Thank you again for your suggestion.

Reviewer 2 Report
The paper reviews the genome-wide association studies conducted in the past 10 years in samples with patients with Alcohol Use Disorders and mood or anxiety disorders.
The authors followed the PRISMA guidelines, and searched (by means of inclusion and exclusion criteria, and certain key words) in several databases.
14 studies were included, and the results support a significant overlap of genetic factors involved in AUD and mood and anxiety disorders.
In the end the authors discuss these results and give some ideas to future research.
The introduction is carefully written and includes relevant and recent literature.
The methods are very well structured and detailed.
No suggestions should be reported.
Author Response

(The authors gave the same response as above.)
